# Role of α2-Adrenoceptor Subtypes in Suppression of L-Type Ca^2+^ Current in Mouse Cardiac Myocytes

**DOI:** 10.3390/ijms22084135

**Published:** 2021-04-16

**Authors:** Edward V. Evdokimovskii, Ryounghoon Jeon, Sungjo Park, Oleg Y. Pimenov, Alexey E. Alekseev

**Affiliations:** 1Institute of Theoretical and Experimental Biophysics, Russian Academy of Science, Institutskaya 3, 142290 Pushchino, Russia; onletaet@gmail.com (E.V.E.); polegiteb@gmail.com (O.Y.P.); 2Department of Cardiovascular Medicine, Center for Regenerative Medicine, Mayo Clinic, Stabile 5, Mayo Clinic, 200 1st Street SW, Rochester, MN 55905, USA; jeon.ryounghoon@mayo.edu (R.J.); park.sungjo@mayo.edu (S.P.)

**Keywords:** G-protein coupled receptors, left ventricle, cell signaling, guanabenz, BRL 44408, ARC 239, JP 1302

## Abstract

Sarcolemmal α2 adrenoceptors (α2-AR), represented by α2A, α2B and α2C isoforms, can safeguard cardiac muscle under sympathoadrenergic surge by governing Ca^2+^ handling and contractility of cardiomyocytes. Cardiomyocyte-specific targeting of α2-AR would provide cardiac muscle-delimited stress control and enhance the efficacy of cardiac malfunction treatments. However, little is known about the specific contribution of the α2-AR subtypes in modulating cardiomyocyte functions. Herein, we analyzed the expression profile of α2A, α2B and α2C subtypes in mouse ventricle and conducted electrophysiological antagonist assay evaluating the contribution of these isoforms to the suppression of L-type Ca^2+^ current (*I*_CaL_). Patch-clamp electro-pharmacological studies revealed that the α2-agonist-induced suppression of *I*_CaL_ involves mainly the α2C, to a lesser extent the α2B, and not the α2A isoforms. RT-qPCR evaluation revealed the presence of *adra2b* and *adra2c* (α2B and α2C isoform genes, respectively), but was unable to identify the expression of *adra2a* (α2A isoform gene) in the mouse left ventricle. Immunoblotting confirmed the presence only of the α2B and the α2C proteins in this tissue. The identified α2-AR isoform-linked regulation of *I*_CaL_ in the mouse ventricle provides an important molecular substrate for the cardioprotective targeting.

## 1. Introduction

Previously, the catalog of myocellular membrane receptors has been expanded to include α2-adrenoceptors (α2-ARs) that in line with other adrenergic receptors (α1- and β-) control the stress-reactive response of cardiomyocytes [1,2]. We have identified that, in addition to the established α2-AR-mediated feedback suppression of sympathetic and adrenal catecholamine release, α2-ARs in cardiac myocytes improve intracellular Ca^2+^ handling and support myocardial contractility [2,3]. The evidence indicates that protective potential of α2-AR in cardiomyocytes can be mobilized not only against the deleterious effects of chronic stimulation by excessive catecholamine but also against angiotensinergic loads to mitigate the development of cardiac dysfunctions [1,4]. In this regard, future therapeutic directions aimed at cardiac specific restoration or enhancement of α2-AR signaling require identifying α2-AR isoforms in ventricular myocytes, which mediate cardioprotective cellular response.

To date, four different α2-AR subtypes have been identified. Mammalian species express α2A, α2B and α2C receptor isoforms encoded by the *adra2A*, *adra2B* and *adra2C* genes, respectively [5,6,7]. Other vertebrates, except crocodiles, also express α2D isoforms encoded by the genes *adra2Da* and *adra2Db* [8,9,10]. In mammals, primarily the α2A- and α2C-receptor subtypes are present in the central neural system, whereas all three receptor isoforms are broadly distributed in peripheral organs. The presynaptic receptor isoforms exhibit different potency to norepinephrine, which is higher for the α2C-AR compared to the α2A-AR, as well as distinct responsiveness to neuronal stimulation frequencies [11,12,13]. Genetic ablations of either α2A or α2C subtypes have allowed discriminating between the α2A-AR-dependent control of plasma norepinephrine and the predominant α2C-AR-driven inhibition of the catecholamine secretion from chromaffin cells. Activation of α2B-AR mediates initial phase of peripheral hypertensive response followed by hypotension that is mediated by α2A-AR. In addition, α2B-ARs in line with other receptor isoforms also mediate the antinociceptive response to α2-AR agonists [14].

Mechanistically, activation of the presynaptic α2-ARs results in suppression of cAMP levels, opening of K^+^ channels and inhibition of voltage-gated Ca^2+^ channels directly affecting the exocytotic machinery [15,16]. Thus, α2-ARs have been recognized as short-loop feedback suppressors of sympathetic and adrenal catecholamine release and, thereby, generally have an inhibitory influence on sympathoadrenergic drive [11,17,18,19]. The range of pharmacology effects of these receptor isoforms in neurons also relies on regulation of other than norepinephrine neurotransmitter release in the central and peripheral nervous, which contributes to anti-depressive potentials of the α2-AR antagonists [20]. Concomitantly, α2-AR agonists are clinically used as an adjuvant for premedication, especially in patients susceptible to preoperative and perioperative stress because of its sedative, anxiolytic, analgesic and sympatholytic profiles [21,22].

The expression of α2A, α2B and α2C in rat hearts was found negligible compared to levels of these receptors in neuronal, kidney, liver or aortic tissues [23], which led to the conventional belief that direct α2-AR-mediated regulation of cardiac excitation or contractile functions must be limited. More recent studies that also identified the expression of all α2-AR isoforms in rat ventricular myocytes [24,25] demonstrated that NO and cGMP were central intracellular messengers mediating α2-AR-signaling [2,25]. Key cardiomyocyte responses to activation of α2-AR include stimulation of endothelial NO synthase (eNOS), reduction of intracellular Ca^2+^ levels and suppression of spontaneous intracellular Ca^2+^ oscillations (presumably through the regulation of SERCA/RyR activities) and inhibition of membrane inward Ca^2+^ currents via L-type Ca^2+^ channels [2,25,26]. Furthermore, by promoting phosphorylation of Erk1/2, Akt and eNOS in left ventricular myocytes, the α2-AR agonist dexmedetomidine improved cardiac recovery after ischemia/reperfusion [24,27]. At the same time, the maladaptive cardiac remodeling associated with development of cardiac hypertrophy and heart failure is accompanied by the functional desensitization/internalization of α2-AR [2,28]. Comparison of the amino acid sequences in the third intracellular loop of α2-AR isoforms, the region responsible for phosphorylation of multiple serine or threonine residues, revealed little sequence homology suggesting the subtype selective desensitization mechanisms [29,30]. Thus, cardiomyocyte-specific targeting of α2-ARs aimed at improving cardiac muscle-delimited stress control demands information about a link between α2-AR subtypes and specific cell-signaling pathways encompassing cardioprotective mechanisms.

While current evidence underlines a cardioprotective potential of α2-ARs in cardiomyocytes, little is known about the specific contribution of α2-AR isoforms in cardiac muscle responses. Herein we probe the expression profile of α2-AR isoforms in the mouse ventricle and analyzed the α2-AR agonist-induced suppression of L-type Ca^2+^ current in isolated cardiomyocytes, using selective antagonists of α2A, α2B and α2C receptor isoforms.

## 2. Results

### 2.1. Activation of α2-ARs Inhibits L-Type Ca^2+^ Current

In isolated mouse cardiomyocytes guanabenz, a non-selective agonist of the α2-AR isoforms, significantly, but reversibly, reduced inward transient currents measured throughout a range of depolarizing membrane potentials (Figure 1a–c). At the applied holding potential (−40 mV) nifedipine (5 µM), a selective inhibitor of L-type Ca^2+^ channels, eliminated the measured inward currents (Figure 1c). This indicates that the measured whole-cell currents represent only the low-threshold voltage-gated L-type Ca^2+^ current (*I*_CaL_) amenable to regulation by guanabenz. Dose-response relationship constructed for the effect of guanabenz on peak values of *I*_CaL_, measured at +10 mV of membrane potential, and fitted with Hill’s equation revealed IC_50_ = 24.8 ± 9.7 µM and *h* = 1.22 ± 0.38 (*n* = 3–5; Figure 1d,e). The time-course of guanabenz-induced inhibition of the peak *I*_CaL_ values demonstrated that the steady-state blocking effect was effectively suppressed by 50 µM of yohimbine, a non-specific antagonist of α2-AR isoforms (Figure 1f). Of note, 50 µM of yohimbine alone induced 23.2 ± 3.1% inhibition of the peak *I*_CaL_ values (*n* = 4; Figure 1f). Suppression of *I*_CaL_ by guanabenz measured in the presence of yohimbine (relative to the effect of this antagonist alone) and fitted by Hill’s equation demonstrated a right-shift of the guanabenz-dependent dose-response curve to IC_50_ = 184.6 ± 41.3 µM and *h* = 1.22 ± 0.51 (*n* = 3–4; Figure 1e). Thus, in mouse cardiomyocytes, the activation of sarcolemmal α2-AR isoforms by guanabenz results in the suppression of *I*_CaL_.

### 2.2. α2A-AR in Guanabenz-Induced Suppression of I_CaL_

To block the α2A receptor isoforms we applied BRL 44404, an established selective antagonist of this adrenoceptor subtype [31]. The agonist guanabenz in the presence of 40 µM of BRL 44408 maintained the suppression of *I*_CaL_ (Figure 2a,b). BRL 44408 alone produced minor inhibition of *I*_CaL_, which was estimated at 9.4 ± 2.2% of control peak *I*_CaL_ values measured at +10 mV (Figure 2b). The antagonist BRL 44408 provided an unessential right shift of guanabenz-dependent dose-response curve, which was fitted by Hill’s equation with IC_50_ = 27.2 ± 4.6 µM and *h* = 1.34 ± 0.28 (*n* = 3; Figure 2c). Therefore, the α2A receptor isoform does not contribute to the suppression of *I*_CaL._

### 2.3. α2B-AR in Guanabenz-Induced Suppression of I_CaL_

To test the role of the α2B isoform in suppression of *I*_CaL_ we used ARC 239, a selective antagonist of this adrenoceptor subtype [32]. ARC 239 at high concentration of 40 µM antagonized the inhibitory effect of guanabenz on *I*_CaL_ only at low doses, and, in contrast to BRL 44404, did not induce detectable effects on *I*_CaL_ when applied alone (Figure 3a,b). Consequently, ARC 239 induced a minor right-shift of the guanabenz dose-response relationship at the agonist levels below 30–40 µM. The Hill’s fitting of the guanabenz dose-response revealed IC_50_ = 33.2 ± 7.2 µM and *h* = 1.51 ± 0.34 (*n* = 3; Figure 3c), indicating a very modest contribution of α2B-ARs in the suppression of *I*_CaL_.

### 2.4. α2C-AR Mediates Guanabenz-Induced Suppression of I_CaL_

To assess the input of the α2C isoform to suppression of *I*_CaL_, the selective antagonist JP 1302 was applied [33]. While the agonistic effects of JP 1302 were not found in competition binding assays or in multiple physiological studies [33], in isolated cardiomyocytes, JP 1302 alone induced significant inhibition of *I*_CaL_ (Figure 4a,b). Our measurements revealed the dose-response relationship for JP 1302–induced *I*_CaL_ inhibition with IC_50_ = 17.9 ± 2.7 µM and *h* = 1.63 ± 0.33 (*n* = 3–5; Figure 4c). When applied at 4 µM, JP 1302 reversed the suppression of *I*_CaL_ induced by 20 μM of guanabenz (Figure 4d,e). The guanabenz dose-response relationships constructed at 4 and 20 μM of JP 1302 indicated significant rightward shifts with IC_50_ = 34.0 ± 2.3 µM, *h* = 1.46 ± 0.13 and IC_50_ = 63.4 ± 4.8 µM, *h* = 1.52 ± 0.15, respectively (*n* = 3–4; Figure 4f). Thus, α2C is the main receptor isoform that in mouse cardiomyocytes mediates guanabenz-induced suppression of *I*_CaL_.

### 2.5. Expression of α2-AR Genes in Mouse Hearts

In the mouse left ventricle, RT-qPCR assay revealed low levels of α2 subtype gene expressions that might explain the necessity for relatively high agonist concentrations to induce the considerable functional output in cardiomyocyte [2,25,26]. This assay did not identify in the mouse left ventricle the expression of *adra2A* mRNA (α2A gene), and it revealed the mean cycle threshold values of 36.8 ± 2.7 (*n* = 3) for *adra2B* (α2B gene) and 33.9 ± 1.8 (*n* = 3) for *adra2C* (α2C gene). The ΔCt values for these genes estimated relative to the *gapdh* mean Ct value of 15.7 ± 0.3 (*n* = 6) indicate a weak expression of both mRNAs, and that the expression levels of *adra2B* were approximately one order of magnitude lower compared to *adra2C* (Figure 5a). In contrast, a more prominent expression of α2-AR isoforms was identified by this assay in the brain lysate with the mean Ct values of 25.6 ± 0.2, 28.9 ± 0.3 and 24.1 ± 0.02 for *adra2A, adra2B* and *adra2C,* respectively, and with the *gapdh* Ct value of 15.9 ± 0.1 (*n* = 3 for all samples). Immunoblotting demonstrated the absence of α2A protein and the presence of α2B and α2C protein subtypes in mouse left ventricle (Figure 5b).

## 3. Discussion

We have established, herein, that in isolated mouse cardiomyocytes guanabenz, an agonist of α2-AR, reversibly suppressed L-type Ca^2+^ currents. Inhibitory analysis using the specific receptor antagonists revealed that the α2-agonist-induced suppression of *I*_CaL_ mainly involves the α2C, to a lesser extent the α2B, and not the α2A receptor isoforms. In general, suppression of *I*_CaL_ in mouse cardiomyocytes in response to activation of α2-ARs is consistent with the data obtained in rat cardiomyocytes [2]. In both species, the intrinsic expression levels of α2-AR genes were low, which may explain the relatively low efficiency of the agonists in cardiomyocytes. However, the profile of α2-AR subtype expressions in the mouse left ventricle was found to be different from expression of corresponding mRNAs in rat cardiomyocytes. In contrast to rat cardiomyocytes, we did not identify the expression of *adra2a* gene in the mouse myocardium. Thus, the input of α2-AR isoforms in diverse cardiomyocyte responses may not be identical among mammals, including humans, necessitating further investigations.

The analysis of protein expression using Western blots confirmed the absence in the left mouse ventricle of the α2A isoform and the presence of the α2B and the α2C proteins. The molecular weight of the Western blot products precipitated by the antibodies against α2B protein in our study (~72 kDa), although heavier than predicted (~50 kDa), exactly correspond to the manufacturer datasheet [34]. In our studies, the Alomone Labs antibody against α2C did not precipitate any protein of the expected molecular weight. However, corresponding Abcam antibodies, in line with the manufacturer specifications, precipitated a 60–70 kDa product in the mouse tissue [35]. Of note, both the Alomone Labs and Abcam specific antibodies against the α2A receptor subtype did not detect the presence of this isoform in the lysate of mouse myocardium tissue. Although there is no clear explanation for the difference in the receptor molecular weights observed here, as well as in the manufacturer assays between mouse and, for instance, human tissues, it is possible that phosphorylation, glycosylation, lipidation, etc., may alternatively contribute to the post-translational modifications of α2-AR proteins in different tissues [36,37,38].

The specific antagonists used here are well-established tools for pharmacological discrimination of α2-AR subtypes in numerous in vivo and in vitro studies [20]. Among other specific antagonists, JP 1302, an antagonist of α2C isoform, exhibited a more significant inhibitory effect on *I*_CaL_ in mouse cardiomyocytes. To the best of our knowledge, the JP 1302-induced suppression of L-type Ca^2+^ currents has not been previously reported. In particular, the antagonism assays employing JP 1302 in hippocampal neurons revealed no effect of this agent alone to synaptic vesicle exocytosis, which depends upon activation of both N-type and P/Q-type voltage-gated Ca^2+^ currents [33,39]. JP 1302 (1–10 µM), in the absence of α2-AR agonists, exhibited no significant effects on contractile force of ventricular strips, but induced a negative inotropic effect in atrial strip samples [40]. In line with the previous studies, our results indicate that the antagonist JP 1302 hardly can be characterized as a partial α2-AR agonist, since, in this case, its effect on *I*_CaL_ would be synergetic to the blocking effect of agonist. In contrast, we identified that JP 1302 at low doses could antagonize the suppression of *I*_CaL_ by guanabenz. However, at present, we cannot specify the alternative α2-AR-independent mechanism by which JP 1302 affects *I*_CaL_.

We believe that the obtained results further underline the significance of α2-AR signaling that by optimizing intracellular Ca^2+^ handling can increase the effectiveness of contractile systolic function and reduce a risk for detrimental Ca^2+^ cellular overload [41]. Furthermore, these data provide an important molecular and genomic basis for understanding the functional reactions of myocardial cells to activation of α2-AR. Such information would be critical for a future development of animal models with a tissue-specific suppression or potentiation of expression of the α2-AR isoforms, as well as for a prospective new gene- or cell-based therapies aimed at treating cardiomyopathy and heart failure [1].

## 4. Materials and Methods

### 4.1. Cell Isolation

Cardiomyocytes were isolated from the left ventricle of isoflurane anesthetized mice (C57BL/6) by enzymatic dissociation [42]. Following open chest surgery, the hearts were exposed, and descending aorta and inferior vena cava were cut. Immediately, 7 mL of HEPES buffer containing (in mM): NaCl, 130; KCl 5; NaH_2_PO_4_, 0.5; Glucose 10; Taurine, 10; Diacetyl monoxime, 10; HEPES 10 (pH 7.8) with 5 mM EDTA was injected into the right ventricle. Next to succeeding ascending aorta clamping, the hearts were perfused via the left ventricle apex by consecutive injections of the following solutions: (i) 10 mL of HEPES buffer + 5 mM EDTA; (ii) 3 mL of HEPES buffer + 1 mM MgCl_2_; (iii) 30–40 mL of HEPES collagenase buffer (Collagenase II, 0.5 mg/mL; Collagenase IV, 0.5 mg/mL; Protease XIV, 0.05 mg/mL) + 1 mM MgCl_2_. The left ventricles were cut into small pieces and isolated cardiomyocytes were obtained by a gentle trituration followed by centrifugation. Proteolytic reactions were quenched by washing with fetal bovine serum. Cells filtered through a 100-μm nylon mesh cell strainer (Thermo Fisher Scientific Inc., Waltham, MA, USA) were resuspended in a low Ca^2+^ media containing (in mM): NaCl, 80; KCl, 10; KH_2_PO_4_, 1.2; MgCl_2_, 5; glucose, 20; taurine, 50; L-arginine, 1; HEPES, 10 (pH 7.35) and underwent gravity settled post-dissociation recovery achieved by stepwise Ca^2+^ increase from 0.2 to 1.8 mM.

### 4.2. Electrophysiology

Membrane currents in isolated cardiac myocytes were measured using the perforated mode of the whole-cell patch clamp technique. Whole-cell membrane potential was controlled through the electrical access obtained by membrane patch perforation induced by amphotericin B (200–250 µg/mL) added to the pipette (4–5 MΩ) containing (in mM): CsCl, 130; MgSO_4_, 5; HEPES, 10 (pH 7.25). The bath solution contained (in mM): NaCl, 80; CaCl_2_, 2; MgSO_4_, 5; KH_2_PO_4_, 1.2; CsCl, 10; tetraethylammonium chloride (TEA-Cl), 20; glucose, 20; L-arginine, 1; HEPES, 10 (pH 7.25). L-type Ca^2+^ currents were elicited by depolarizing rectangular pulses from a holding potential of −40 mV, chosen to inactivate low threshold voltage-gated channels. Currents were measured by using an Axopatch 200B amplifier (Molecular Devices, San Jose, CA, USA). Progress of membrane perforation was monitored online by estimation of serious resistance and cellular capacitance values based on analysis of capacitive transient currents. Protocol of stimulation, determination of cell parameters and data acquisition were performed by using the custom BioQuest software and a L-154 AD/DA converter (L-card, Moscow, Russia) [43]. Following formation of the perforated whole-cell patch-clamp configuration, approximately 20 MΩ of series resistances was compensated by 80–100% (at the LAG control 80–100 µs) to final values less than 12 MΩ. Measurements were performed at 31 ± 0.5 °C, using a HCC-100A temperature controller (Dagan Corp., Minneapolis, MN, USA). Peak current values measured at +10 mV of membrane potentials were normalized to the cell capacitances and presented in graphs as “Peak *I*_CaL_ density”.

### 4.3. RNA Isolation and RT-qPCR Assay

Total RNA was extracted from left ventricular walls (~100 mg) isolated from isofluorane (2%)-anesthetized mice. Tissue samples were stored and homogenized in 1 mL of ExtractRNA reagent (BC032 Evrogen, Moscow, Russia). RNA isolation was performed according to the manufacturer recommendations. At the final step, RNA was precipitated by 3 volumes of 96% ethanol. The pellet, following washing in cold 75% ethanol and air-drying, was resuspended in 100 µL of deionized water. RNA preps were treated with DNAse I (04716728001 Roche) for 1 h at 37 °C, followed by DNAse inactivation for 10 min at 70 °C, to reduce genomic DNA contamination. The quality of RNA isolation was controlled spectrophotometrically, using the NanoDrop 2000c (Thermo Fisher Scientific Inc., Waltham, MA, USA). Synthesis of cDNA was carried out with the commercially available reverse transcription MMLV RT kit (Evrogen, Moscow, Russia) using an oliogo(dT)_18_ primer. Then, qPCR for each α2-AR isoform was performed with the Applied Biosystems^TM^ 7500 Real-Time PCR System (Thermo Fisher Scientific Inc., Waltham, MA, USA), using HS Taq DNA polymerase (PK018, Evrogen, Moscow, Russia) according to the following protocol: 94 °C for 5 min, 40 cycles of 94 °C for 20 s and then 60 °C for 1 min. The expression levels of particular *adra2* gene of interest (*GOI*) were expressed relative to the expression of housekeeping gene *gapdh*, as following: ΔCt = Ct[*gapdh*]—Ct[*GOI*]. Primers (forward, …_F; reversed, …_R and probe, …_P) used for the assay are listed in Table 1.

### 4.4. Western Blot

Proteins were separated in 10% SDS-PAGE, transferred to nitrocellulose membranes (sc-3724, 0.45 µm, Santa Cruz Biotechnology) and probed with antibodies against α2A (AAR-020, Alomone labs, lot AAR020AN0202), α2B (AAR-021, Alomone labs, lot AAR021AN0202) and α2C (ab46536, Abcam) diluted to 1:100. Counterstain was performed with horseradish peroxidase (HRP)-conjugated anti-rabbit (Santa Cruz, sc-2004, 1:200 dilution) secondary antibodies. HRP signals were detected by using 3,3′-diaminobenzidine tetrahydrochloride (DAB) substrate (Amresco, E733). All expression assays were performed in triplicate in at least two independent experiments.

### 4.5. Drugs

The stock solutions (10–20 mM) of the α2-AR agonist guanabenz acetate (Sigma, St. Louis, MO, USA), non-selective antagonist yohimbine dihydrochloride (Sigma) and selective antagonists BRL44408 maleate (α2A-AR), ARC239 dihydrochloride (α2B-AR) and JP1302 dihydrochloride (α2C-AR), all from Tocris, were prepared in deionized water.

### 4.6. Data Analysis and Presentation

The averaged data are presented as mean ±SEM. Inhibition profiles of measured membrane currents in dose-response experiments were presented relative to the control current values measured before drag application and fitted with corresponding Hill’s equations:Relative inhibition = [1 + (*c*/*IC_50_*)]^−*h*^,
where *IC_50_* is the half-inhibition constant, *c* is the concentration and *h* is the Hill coefficient.

## Figures and Tables

**Figure 1 ijms-22-04135-f001:**
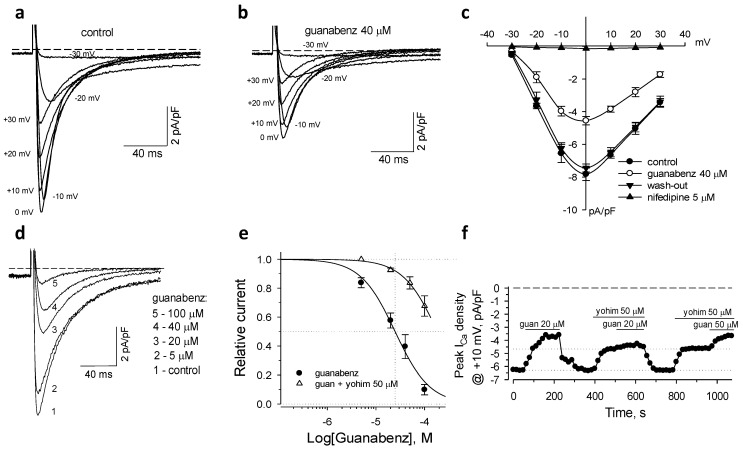
Guanabenz, an agonist of α2-AR, suppressed L-type Ca^2+^ currents (*I*_CaL_) in isolated mouse cardiac myocytes. (**a**,**b**) Whole-cell currents were measured in response to depolarization from −40 mV to membrane potentials indicated nearby the current traces. (**c**) Voltage–current relationships demonstrated reversible guanabenz-induced inhibition of membrane currents that, under the experimental conditions, entirely belong to the dihydropyridine (nifedipine)-sensitive *I*_CaL_ (*n* = 3–5). (**d**) Representative dose-dependent suppression of *I*_CaL_ by guanabenz. (**e**) Dose-dependent relationships of the guanabenz-induced suppression of *I*_CaL_ constructed based on the peak *I*_CaL_ values relative to the control peak *I*_CaL_. Yohimbine, a non-specific antagonist of α2-AR isoforms, induced a rightward shift of the guanabenz-dependent dose effect. This relationship was constructed relative to the values measured in the presence of yohimbine alone. Curves represent the Hill’s fits with parameters indicated in the text. (**f**) Representative time-course of the peak *I*_CaL_ density values in the presence of guanabenz (guan) and yohimbine (yohim). Horizontal bars denote the protocol of the drug applications.

**Figure 2 ijms-22-04135-f002:**
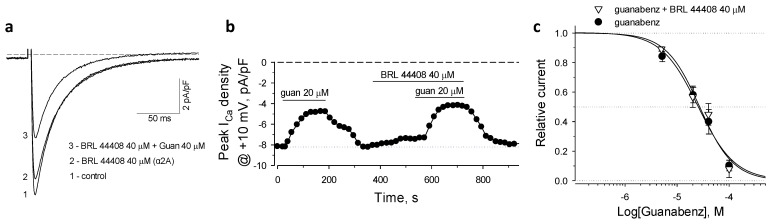
BRL 44408, a specific antagonist of the α2A isoform, did not affect the guanabenz-induced suppression of *I*_CaL_. (**a**,**b**) Representative *I*_CaL_ recordings and time-course of peak *I*_CaL_ density values in the presence of guanabenz (guan) and BRL 44408. Horizontal bars denote the protocol of the drug applications. (**c**) BRL 44408 did not induce a significant rightward shift of dose-dependent inhibition of *I*_CaL_ by guanabenz. Curves represent the Hill’s fits with parameters indicated in the text.

**Figure 3 ijms-22-04135-f003:**
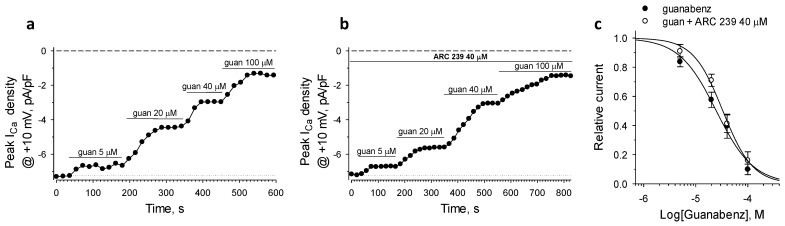
The effect of ARC 239, a specific antagonist of the α2B isoform, on the guanabenz-induced suppression of *I*_CaL_. (**a**,**b**) Representative time-courses of peak *I*_CaL_ density values of guanabenz (guan)-induced suppression of *I*_CaL_ in the absence and presence of the antagonist ARC 239. Horizontal bars denote the protocol of the drug applications. (**c**) ARC 239 induced a minor rightward shift of dose-dependent inhibition of *I*_CaL_ at the lower concentrations of guanabenz.

**Figure 4 ijms-22-04135-f004:**
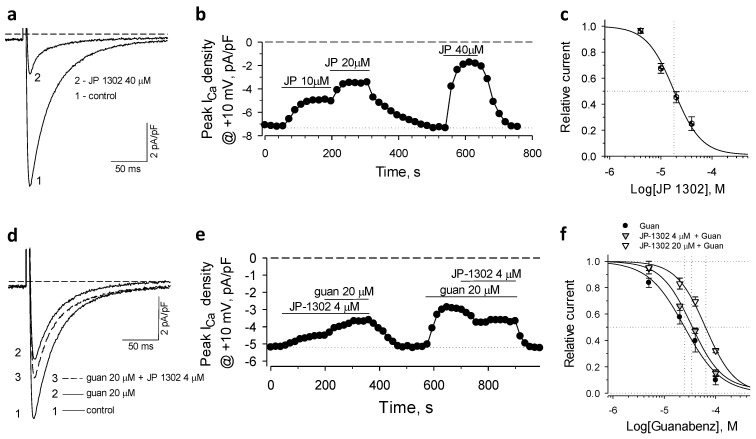
The effect of JP 1302, a specific antagonist of the α2C isoform, on the guanabenz-induced suppression of *I*_CaL_. (**a**,**b**) Representative *I*_CaL_ recordings and time-course of peak *I*_CaL_ density values measured in the presence of JP 1302 alone. (**c**) dose-response relationship of JP 1302-induced suppression of *I*_CaL_ (the Hill’s plot parameters in the text). (**d**,**e**) Representative *I*_CaL_ recordings and time-course of peak *I*_CaL_ density values obtained at low concentration of JP 1302 that reversed guanabenz-induced suppression of *I*_CaL_. (**f**) JP 1302 induced rightward shifts of dose-dependent inhibition of *I*_CaL_ by guanabenz (the Hill’s plot parameters in the text). The relationships in the presence of JP 1302 were constructed relative to the current values measured in the presence of the antagonist alone.

**Figure 5 ijms-22-04135-f005:**
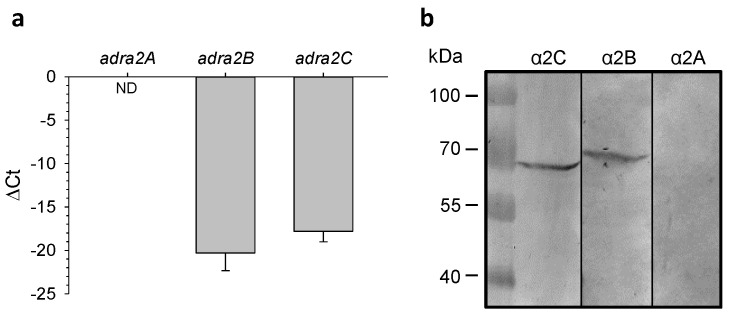
The mRNA and protein expression levels of the α2-AR isoforms in the mouse left ventricle. (**a**) mRNA expression of the α2 adrenoceptor genes obtained using the RT-qPCR assay. In contrast to *adra2B* and *adrs2C* genes the expression of *adra2A* was not detected (ND). (**b**) Western blots confirmed the expression of α2B and α2C but not α2A receptor proteins in the mouse left ventricular tissue.

**Table 1 ijms-22-04135-t001:** Primers for RT-qPCR assay.

Name	Primer Sequence	GenBank Index	Localization
Adra2a_F	TTTCCCCTGTGCCTAACTGC	NM_007417.5	3072–3091
Adra2a_R	TGGCTTTATACACGGGGCTG		2250–2269
Adra2a_P	FAM-ACAGCGATGGACCAAGGCAGAAGG-BHQ1		2222–2246
Adra2b_F	TTCAACCTCGCAGAGAGCAG	NM_009633.4	2728–2747
Adra2b_R	CTCTAGCGCATTTCCCCCAT		2834–2815
Adra2b_P	GCCTGCCGCCT-R6G-ACTTGCAGCAGGG-BHQ1		2758–2781
Adra2c_F	AGTTGCCAGAACCGCTCTTT	NM_007418.3	2546–2565
Adra2c_R	GAGCGCCTGAAGTCCTGATT		2648–2629
Adra2c_P	Cy3-TGCAACAGTTCGCTCAACCCGGT-BHQ2		2590–2612
GAPDH_F	GGGTCCCAGCTTAGGTTCAT	NM_001289726.1	32–51
GAPDH_R	CCCAATACGGCCAAATCCGT		131–112
GAPDH_P	Cy5-CAGGAGAGTGTTTCCTCGTCCCGT-BHQ2		62–85

## Data Availability

The data presented in this study are available in the article.

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
