# Peer review of "Role of α2-Adrenoceptor Subtypes in Suppression of L-Type Ca2+ Current in Mouse Cardiac Myocytes"

_ijms, 2021, doi:10.3390/ijms22084135_

Round 1

Reviewer 1 Report

Major concerns pertain to the direct quantification of expression of all three isoforms in mouse ventricular tissue:

A. qPCR analysis: just presenting ΔCt values is not adequate; depending on the Ct value for Gapdh, ΔCt values of negative 18-20 suggest Ct values for GOI in the high 30s or 40s. It is difficult to accept such high Ct values as reliable; readers/reviewers need to be provided the direct Ct values for these mRNAs/cDNAs.

B. Were -RT (first-strand cDNA synthesis without inclusion of reverse transcriptase) controls included in the qPCR analyses?

C. What positive controls were included in the qPCR analyses? How does expression in mouse ventricles compare with other tissues?

D. Given such exceedingly low mRNA levels, it is surprising that protein bands could even be detected in immunoblots, especially using  chemiluminescence technology. How much protein was loaded per lane?

E. Immunoblots should be repeated with positive controls, especially given discrepancies in molecular weight.

F. As group has skills to isolate cardiomyocytes, why were cardiomyocytes used as source of total RNA/protein instead of whole ventricular tissue? This would help clarify concerns regarding reliability of results shown.

Minor:
a. In lines 182/186, Fig. 5 is cited incorrectly as Fig. 6.

Author Response

Please, see our response in the attached Word file.

Reviewer 2 Report

The authors of the present paper analyzed the expression profile of the sarcolemmal α2 adrenoceptors (α2-AR) isoforms, α2A, α2B and α2C, in mouse ventricle. Then, they performed electro-pharmacological assays in order to assess the contribution of these isoforms in the suppression of L-type Ca2+ current (ICaL). The main findings of this study are the detection of the α2B and α2C but not α2А isoform genes in the mouse left ventricle, and the involvement of  α2С and α2В, but not the α2А isoforms in the α2-agonist-induced suppression of ICaL. Scientifically, the present study is of a great interest to the field. However, it contains some flaws that should be addressed:

Point 1:

-The English language should be revised by a native English speaker. Many sentences lack coherence (eg. 1st sentence of the abstract). Please avoid complex and long sentences. 

Point 2:

-Sentence 211: the difference between the two commercialized antibodies is not well explained. Please clarify and reformulate the sentence.

Point 3:

-Material and methods/ Cell isolation: Please add a declaration whether the study comply with the ethical protocols in terms of experimental animal manipulation.

Point 4:

-The discussion section should be further developed involving more references. 

-Please use more recent references in your discussion. Most of the cited papers are from the nineties. 

Point 5:

-The authors haven't mentioned which statistical test they have used to evaluate the significance of of results. pvalues should be presented when appropriate.

-The authors haven't mentioned how many times the expression evaluation assays experiments have been repeated.

Minor comments: 

Line 204: "in the muse myocardium"-->"in the mouse myocardium. 

Line 248: "anesthetize"--> "anesthetized"

Line 303: "Implemented"--> "used"

Thank you

Author Response

We thank the Reviewers for their credits to our study and critiques that guided us through revision of the manuscript. Following all recommendations, we extended the article with validation of the isoforms expression, corrected misspellings and improve the text readability.

Reviewer #2

Point 1:

-The English language should be revised by a native English speaker. Many sentences lack coherence (eg. 1st sentence of the abstract). Please avoid complex and long sentences. 

We revised our manuscript according to the Reviewer’s critique. Specifically, we changed 1st sentence of the Abstract as follows: “Sarcolemmal α2 adrenoceptors (α2-AR), represented by α2А, α2B and α2C isoforms, by governing Ca2+ handling and contractility of cardiomyocytes can safeguard cardiac muscle under sympathoadrenergic surge.”

Point2:

-Sentence 211: the difference between the two commercialized antibodies is not well explained. Please clarify and reformulate the sentence.

The unclear sentence in line 211 we revised as follows (line 209 in the amended version): “In our studies, the Alomone Labs antibody against α2C did not precipitate any protein of the expected molecular weight. However, corresponding Abcam antibodies, in line with the manufacturer specifications, precipitated a 60  ̶ 70 kDa product in the mouse tissue [35].” Of note, as we stated in line 212, both the Alomone Labs and Abcam antibodies against the α2A receptor did not precipitate any receptor isoform in the lysate of mouse myocardium tissue.

Point 3:

-Material and methods/ Cell isolation: Please add a declaration whether the study comply with the ethical protocols in terms of experimental animal manipulation.

In the manuscript template, provided by the Editorial Board of IJMS, this statement was placed in the Institutional Review Board Statement following the Methods section (see line 329).

Point 4:

-The discussion section should be further developed involving more references. 

-Please use more recent references in your discussion. Most of the cited papers are from the nineties. 

In response to the critique, we extended the list of citations by introducing new references [20], [36-38], [42], in the Discussion and replaced references [15], [16] in the Introduction.

We would like to argue the Reviewer’s comment regarding “the most cited papers form nineties”. Our manuscript contains 13 papers published before 2000, which represents only 30% of references. Indeed, we prefer to cite the most original publications and believe that scientific data have no expiration date, unless they could be wrong or incomplete. Therefore, it does not matter when the data were obtained: 5 or 50 years ago if they are still actual. Nonetheless, with respect to the Reviewer’s opinion we extended our References with the papers published within the last decade.

Point 5:

-The authors haven't mentioned which statistical test they have used to evaluate the significance of of results. pvalues should be presented when appropriate.

In the Method, section 4.6. Data analysis and presentation, we specified the statistical analysis aimed at documenting antagonistic effects on the Ca2+ current suppression. This well-established regression analysis is based on the fitting experimental data by the parameters of Hill’s equation. The parameters obtained from fitting each dose-response are present throughout the Results as mean ± SEM. Although, the obtained parameters were presented in form of random variables, they do not characterize a distribution of experimental population. Rather they indicate the quality of the data fitting and, thereby, hardly can be used for a statistical significance testing. Therefore, we did not specify any null hypothesis, the significance of which can be tested using, for instance, the analysis of variance. We also would like to note that statistical significance might not always underscore the scientific significance. In our study, the scientific significance of specific receptor isoforms in the suppression of ICaL was validated combining the Hill’s regression results and the mRNA/protein expression analysis.

-The authors haven't mentioned how many times the expression evaluation assays experiments have been repeated.

All expression assays were performed in triplicate in at least two independent experiments, as now we stated in line 309.

Minor Comments:

Line 204: "in the muse myocardium"-->"in the mouse myocardium. 

corrected

Line 248: "anesthetize"--> "anesthetized"

corrected

Line 303: "Implemented"--> "used"

replaced

Round 2

Reviewer 1 Report

1. Ct values in the low 30s are an unreliable basis to use as proof of expression of any gene/protein. It might help strengthen data is cDNA from brain and/or other tissues were included in the same assays to get a comparative account.

2. Recommendation in point 1. would also help assess reliability of immunoblot comparing protein expression of alpha2b in brain and heart. 

Author Response

We thank the Reviewer again for his/her constructive critiques and recommendations. In response, we conducted additional RT-qPCR assay in the brain tissue and incorporated obtained results with the following sentence (line 183):

“In contrast, a more prominent expression of α2-AR isoforms was identified by this assay in the brain lysate with the mean Ct values of 25.6 ± 0.2, 28.9 ± 0.3 and 24.1 ± 0.02 for adra2A, adra2B and adra2C, respectively, and with the gapdh Ct value of 15.9 ± 0.1 (n = 3 for all samples).”

Reviewer 2 Report

It is obvious that the manuscript has been improved however, it still needs some text editing. There are several spelling errors and some sentences need reformulation. Examples are cited below:

  • First sentence in the abstract: This sentence still needs to be reformulated. I would suggest the following: "Sarcolemmal α2 adrenoceptors (α2-AR), represented by α2А, α2B and α2C isoforms, can safeguard cardiac muscle under sympathoadrenergic surge by 
    governing Ca2+ handling and contractility of cardiomyocytes."
  • Line 29: "Previously,...." 
  • Line 29: "membrane receptors have has been...." (as it refers to the catalog).
  • Line 129: "maintained suppressing the suppression ....."
  • Line 134: "contribute to......."
  • Line 181: "reviled..." do you mean revealed?
  • Line 211: "wight..." do you mean "weight...."
  • Line 213-216: please review this sentence. Something is missing there.
  • Line 236: "In line with the previous studies, ...."
  • Line 244: "rick..." do you mean risk?

Author Response

We thank the Reviewer for the careful reading and suggestions allowing improvement of our manuscript. We incorporated in the text all recommended corrections.